# Amino Acids in Rice Grains and Their Regulation by Polyamines and Phytohormones

**DOI:** 10.3390/plants11121581

**Published:** 2022-06-15

**Authors:** Jianchang Yang, Yujiao Zhou, Yi Jiang

**Affiliations:** Jiangsu Key Laboratory of Crop Genetics and Physiology, Co-Innovation Center for Modern Production Technology of Grain Crops, Yangzhou University, Yangzhou 225009, China; zhouyujiao1028@163.com (Y.Z.); jianyiedu@163.com (Y.J.)

**Keywords:** amino acids, anabolism, crop management, phytohormones, rice, spatiotemporal distribution

## Abstract

Rice is one of the most important food crops in the world, and amino acids in rice grains are major nutrition sources for the people in countries where rice is the staple food. Phytohormones and plant growth regulators play vital roles in regulating the biosynthesis of amino acids in plants. This paper reviewed the content and compositions of amino acids and their distribution in different parts of ripe rice grains, and the biosynthesis and metabolism of amino acids and their regulation by polyamines (PAs) and phytohormones in filling grains, with a focus on the roles of higher PAs (spermidine and spermine), ethylene, and brassinosteroids (BRs) in this regulation. Recent studies have shown that higher PAs and BRs (24-epibrassinolide and 28-homobrassinolide) play positive roles in mediating the biosynthesis of amino acids in rice grains, mainly by enhancing the activities of the enzymes involved in amino acid biosynthesis and sucrose-to-starch conversion and maintaining redox homeostasis. In contrast, ethylene may impede amino acid biosynthesis by inhibiting the activities of the enzymes involved in amino acid biosynthesis and elevating reactive oxygen species. Further research is needed to unravel the temporal and spatial distribution characteristics of the content and compositions of amino acids in the filling grain and their relationship with the content and compositions of amino acids in different parts of a ripe grain, to elucidate the cross-talk between or among phytohormones in mediating the anabolism of amino acids, and to establish the regulation techniques for promoting the biosynthesis of amino acids in rice grains.

## 1. Introduction

Amino acids are not only basic constituent units of proteins, but also act as metabolites/metabolic intermediates, enhancing plant growth and development as well as responding to various stresses [1,2]. Amino acids in the grains of cereal crops are important nutrition sources for both humans and animals [3,4,5]. Malnutrition caused by the insufficient intake of amino acids, especially the essential amino acids that humans and livestock are unable to carry out by de novo biosynthesis, is a serious problem around the world. Worldwide, it is estimated that more than a third of all deaths among children under the age of five are linked to protein or amino acid malnutrition [6]. In 2017, there were 515 million people with chronic malnutrition due to the insufficient intake of amino acids around the world, and these people are mainly concentrated in developing countries and regions with rice (*Oryza sativa* L.) as their staple food [7,8]. Rice is one of the most important food crops in the world, and more than 50% of the world’s population lives on rice [9,10,11]. Exploring regulatory ways to increase the content of amino acids, especially essential amino acids, in rice grains is an effective approach to meeting nutrition requirements for these populations.

There are reports showing that the content and compositions of amino acids in a ripe rice grain (not including palea and lemma) and their effectiveness in the human diet vary greatly with the parts of a grain [12,13,14]. The biosynthesis and metabolism of amino acids in the grain during filling (filling grains) are complex processes catalyzed by many enzymes, and properly regulating the process could increase the content of amino acids, especially essential amino acids such as lysine (Lys) in a ripe grain [14,15,16]. It is also generally proposed that phytohormones and plant growth regulators play vital roles in regulating plant growth, development, grain yield and quality, and include the biosynthesis of amino acids in cereals [17,18,19,20]. Therefore, this paper aims to review (1) the content and compositions of amino acids and their distribution in different parts of ripe rice grains, (2) the biosynthesis and metabolism of amino acids in filling grains, and (3) the roles of phytohormones and plant growth regulators in regulating the anabolism of amino acids in grains during filling, with a focus on the roles of polyamines (PAs), ethylene, and brassinosteroids (BRs) in this regulation. The focus of future research on increasing the content of amino acids in rice grains is also discussed. This review helps in further understanding the mechanism underlying the formation of amino acids and exploring regulatory ways to enhance the biosynthesis of amino acids, especially essential amino acids, in rice grains.

## 2. Content, Compositions, Anabolism of Amino Acids in Rice Grains

### 2.1. Content, Compositions, and Distributions of Amino Acids in the Ripe Grain

In the ripe grain (caryopsis) of rice, there are 20 main amino acids, including eight essential amino acids that cannot be synthesized by humans and animals, namely Lys, threonine (Thr), methionine (Met), tryptophan (Trp), phenylalanine (Phe), isoleucine (Ile), leucine (Leu), and valine (Val) [2,5,21]. Among them, Lys, Thr, and Met are considered as the first, second, and third most restrictive essential amino acids in rice, respectively. The lack of any of these three amino acids in humans or animals will significantly reduce the absorption and utilization of other amino acids and may result in serious diseases [2,7,22]. For example, Lys deficiency can decrease the defense ability of mammalian cells against viruses and cause osteoporosis and hyperlysinemia in humans [22,23,24]. Lack of Met can result in methylation-related disorders such as tumorigenesis, neurological disorders, and atherosclerosis [25,26,27]. Insufficient intake of Thr can lead to insanity, dyspepsia, and fatty liver [2,28,29]. The content of various amino acids, especially the content of essential amino acids, determines the nutritional value of proteins in the grain, especially Lys content, which is a main index to judge the nutritional quality of rice [21,30,31]. There are reports showing that adding 0.2% Lys and 0.1% Thr to milled rice grains could significantly improve the quality of protein in feeding animals and increase their growth rate [2,32,33]. It is speculated that if the Lys content in rice grains is increased by 20–30%, that is, from 0.25 g g^−1^ nitrogen (N) to 0.32 g^−1^ N, bringing it close to the 0.34 g^−1^ N that is recommended by the World Health Organization, it will play an important role in enhancing people’s physique, especially in improving the nutritional level of people in the countries where rice is the staple food [2,32,33,34].

The amino acids in ripe rice grains are mainly protein-bound amino acids and the proteins are mainly storage proteins [12,35,36]. The storage proteins include albumin, globulin, prolamin, and glutenin, which are soluble in water, in salts, in alcohol, and in acidic or basic solutions, respectively [36,37,38]. The content, composition, and distribution of amino acids in ripe grains vary greatly with the types of storage proteins that are located in different parts of a grain. Generally, a ripe rice grain (caryopsis) is composed of a pericarp, including the epicarp, mesocarp and endocarp; a testa; a nucellar layer; an aleurone layer; an endosperm; and an embryo (Figure 1), and amino acids in albumin and globulin are mainly distributed in the pericarp, testa, nucellar layer, and aleurone layer, while those in prolamin and glutenin are mainly distributed in the endosperm [36,37,38,39,40]. To improve palatability and digestion of rice, ripe grains (caryopses) need processing, in which the pericarp, testa, nucellar layer, aleurone layer, and embryo are removed by a polishing machine. Although albumin and globulin contain high contents of essential amino acids such as Lys and Thr, they are almost completely removed during processing and cannot be eaten by people [34,36,37,38]. Glutenin and prolamin in the endosperm are mainly accumulated in the protein body II (PB-II) or the protein body I (PB-I) [34,35,36,37,38]. Glutenin is composed of 37–39 KD acidic subunits and 20–22 KD basic subunits, which are bound together by a disulfide bond and stored in PB-II, and prolamin is composed of 13 KD subunits, and its polypeptide is synthesized and directly assembled into PB-I in the endoplasmic reticulum [35,36,37,38]. In PB-I, the contents of tyrosine (Tyr) and Trp in prolamin are high, while the contents of essential amino acids such as Lys, Thr, and Met are very low, so its nutritional value is low. In contrast, glutenin stored in PB-II is rich in essential amino acids such as Lys, Thr, Ile, and Leu, which are termed high-quality proteins [37,38,41]. Raising varieties with a high glutenin (low prolamin) content to increase the content of essential amino acids such as Lys in the grains has become an important goal of good-quality breeding in rice [37,38,41,42].

In addition to protein-bound amino acids, there are free amino acids in ripe grains of rice [21,42]. Although the content of free amino acids in a ripe grain is very low, approximately 0.35–0.55% of the total amino acids [21,43], free amino acids are the substrate of protein synthesis, and their content and compositions play an important role in regulating the quantity and quality of proteins during grain maturation [42,43,44,45]. Moreover, the content and compositions of free amino acids in ripe grains have an important impact on the appearance and taste quality of rice [45,46,47]. Increasing the content of free amino acids in the ripe grain, especially the content of glutamate (Glu), is conducive to improving the taste quality of rice [45,46,47].

The content and compositions of amino acids in rice grains may vary among grains located at different positions on the panicle. In general, the early-flowering grains located at the upper and the middle parts of a rice panicle are termed superior grains because of their fast grain filling, good plumpness, and heavy grain weight, and the late-flowering grains located at the lower part of a panicle are termed inferior grains due to their slow filling, poor plumpness, and small grain weight [48,49,50]. The differences in grain filling (the process of starch synthesis and accumulation in grains) between superior and inferior grains in cereals and their mechanism and regulation approaches are well documented [51,52,53,54,55]. However, research on the differences in the increase rate of amino acids in filling grains and contents of amino acids in milled grains between superior and inferior grains is very limited. A study on a *Japonica* rice cultivar showed that, in comparison with inferior grains, superior grains exhibited a higher maximum increase rate and mean increase rate for both total essential and total nonessential amino acids during the grain-filling period, and reached the maximum increase rate earlier, and the mean and the maximum increase rates were closely associated with the content of amino acids in a ripe grain [56]. However, the mechanism underlying the difference in the contents of amino acids between superior and inferior grains remains unclear.

### 2.2. Biosynthesis and Metabolism of Amino Acids in the Filling Grain

In cereals such as rice and wheat (*Triticum aestivum* L.), glutamine (Gln) is the main form of N molecule transported from vegetative organs (leaves, leaf sheaths, stems, etc.) to grains [57,58]. The transfer of NH_4_^+^ between Gln and α-ketoglutarate is regulated by the cyclic catalysis of glutamine synthetase (GS, EC 6.3.1.2) and glutamate synthase (GOGAT, EC 1.4.7.1) to produce Glu; Glu is converted into aspartate (Asp), alanine (Ala), and other amino acids under the catalysis of aminotransferases such as glutamic–oxaloacetic transaminase (GOT, EC 2.6.1.1, also known as aspartate transaminase, AST) and glutamic pyruvic transaminase (GPT, EC 2.6.1.2, also known as alanine aminotransferase, ALT) [58,59,60]. It is reported that the 20 main amino acids in rice grains can be divided into six families: aspartate, glutamate, alanine, serine (Ser), aromatic, and heterocyclic amino acids (histidine), according to their anabolic pathways or carbon skeleton components [58,59,60]. Among them, the aspartate family contains Asp and four essential amino acids for human beings, including the first, second, and third most restrictive essential amino acids, Lys, Thr, and Met, and the essential amino acid Ile (Figure 2). Therefore, the anabolism of amino acids, especially Lys, in the aspartate family has always been the research focus of amino acid biosynthesis and metabolism in rice grains [58,59,60,61].

It is clear that aspartate kinase (AK, EC 2.7.2.4) is the first key enzyme in amino acid biosynthesis of the aspartate family, regulating the biosynthesis of Lys, Thr, Ile, and Met; dihydropicolinate synthase (DHPS, EC 4.2.1.52) is a major rate-limiting enzyme in the branch pathway of Lys synthesis, and enhancing its activity can increase Lys synthesis; Lys α-ketoglutarate reductase (LKR; EC 1.5.1.8, also known as Lys 2-oxoglutarate reductase, LOR) and saccharopine dehydrogenase (SDH, EC 1.5.1.9) are bifunctional Lys-degrading enzymes, and inhibiting LKR/SDH activity or their gene expression can significantly increase Lys content in rice and maize (*Zea mays* L.) grains [2,45,59,60,61]. In addition, homoserine kinase (HK, EC 2.7.1.39), threonine synthase (TS, EC 4.2.99.2), cysthationine ɤ-synthase (CS, EC 4.9.99.9), and threonine dehydrogenase (TD, EC 4.2.1.16) also play important roles in the biosynthesis and metabolism of amino acids of aspartate family (Figure 2). Enhancing the activities of these enzymes or the expression of the genes encoding these enzymes could increase the contents of the essential amino acids Thr, Met, and Ile [58,59,60,61].

## 3. Roles of PAs and Phytohormones in Regulating Anabolism of Amino Acids in Rice

### 3.1. Roles of PAs and Ethylene

PAs and ethylene are important plant growth regulators mediating many physiological processes such as cell division, seed growth and development, senescence, and responses to environmental stresses [62,63,64]. In higher plants, the major PAs are putrescine (Put, diamine), spermidine (Spd, triamine), and spermine (Spm, Tetraamine). Put can be directly synthesized from ornithine via ornithine decarboxylase (ODC, EC 4.1.1.17) or indirectly from arginine via arginine decarboxylase (ADC, EC 4.1.1.19), and Spd and Spm are synthesized via Spd synthase (EC 2.5.1.16) and Spm synthase (EC 2.5.1.22), respectively, by the sequential addition of aminopropyl groups to Put (Figure 3). The aminopropyl groups are generated from S-adenosyl-L-methionine (SAM) by SAM decarboxylase (SAMDC, EC 4.1.1.50) [64,65,66,67,68]. SAM is also a precursor of ethylene, which is produced from SAM via 1-aminocylopropane-1-carboxylic acid (ACC) by ACC synthase (EC 4.4.1.14) and ACC oxidase (EC 1.4.3) (Figure 3).

High levels of PAs (Spd and Spm) have been observed to be closely associated with a higher kernel set and better seed development in crops [69,70,71]. It is reported that higher PAs (Spd and Spm) could accelerate endosperm cell division and grain filling in rice by enhancing activities of the enzymes involved in sucrose-to-starch conversion in the developing rice endosperm, such as sucrose synthase (SuSase, EC 2.4.1.13), ADP glucose pyrophosphorylase (ADPGP, EC 2.7.7.27), starch synthase (StSase, EC 2.4.1.21), and starch branching enzyme (SBE, EC 2.4.1.18) [62,66]. Recent work [72] showed that higher PAs play vital roles in modulating the biosynthesis of amino acids in rice grains, with several pieces of evidence. (1) Rice cultivars (either *Japonica* or *Indica*) with higher concentrations of free Spd and free Spm in the grains during filling exhibited higher content of amino acids in the milled grains than those with lower PA concentrations, and the concentrations of free Spd and free Spm in the filling grains are significantly and positively correlated with the contents of total essential amino acids, total nonessential amino acids, and total amino acids in the milled grain [72]. (2) Application of Spd or Spm to rice panicles at the early grain-filling stage significantly increased the activities of enzymes involved in the biosynthesis of amino acids in the grain including GS, GOGAT, AST, AK, and AST during grain filling, and accordingly, significantly increased the content of amino acids in milled grain of rice, and applying methylglyoxal-bis (guanylhydrazone) (MGBG, an inhibitor of Spd and Spm synthesis that inhibits SAMDC activity) had the opposite effect [72]. (3) Adoption of moderate soil-drying, or alternate wetting and moderate soil-drying (AWMD), or proper application of N fertilizer significantly increased the concentrations of PAs (Spd and Spm) by enhancing the activities of enzymes in PA synthesis, such as SAMDC, Spd synthase, and Spm synthase, in the grain during the filling, and consequently significantly promoted grain filling and increased the content of amino acids in milled grain of rice [62,73]. Furthermore, Spd and Spm levels in rice root exudates during grain filling have been observed to be significantly and positively correlated with the contents of glutenin, total proteins, and amino acids in the milled grain of rice, and PAs are proposed to promote the uptake and transport of amino acids by enhancing the activity of amino acid permease and the expression of genes encoding this protein [68,74]. In addition, PAs are observed to enhance grain filling and amino acid biosynthesis by means of the scavenging of reactive oxygen species (ROS) [62,66,75]. It should be noted, however, that the concentration of diamine Put in rice grains was observed to be significantly and negatively correlated with grain-filling rate, and was not significantly correlated with the content of amino acids in the milled grain of rice [66,72]. Applying Put to rice panicles at the early grain-filling stage had no significant effect on the activities of the enzymes related to amino acid biosynthesis and on the content of amino acids in the grain [72]. It is presumed that the physiological activity of Put in rice grains is very low and an excessive accumulation of Put in plant organs may inhibit growth and development, including amino acid biosynthesis [66,72,75].

In contrast to PAs (Spd and Spm), ethylene plays a role in impeding grain filling and biosynthesis of amino acids in rice grains [76,77,78]. There are three pieces of evidence for this: (1) either *Japonica* or *Indica* rice cultivars with a greater ethylene evolution rate or higher ACC concentrations in the filling grain had a smaller grain-filling rate and lower amino acid content than those with a smaller ethylene evolution rate or lower ACC concentrations, and the ethylene evolution rate or ACC concentration in the grain was significantly and negatively correlated with grain-filling rate or with amino acid content [76,77,78]; (2) the application of ethephon (an ethylene-releasing agent) or ACC to rice panicles markedly reduced, whereas applying aminoethoxyvinylglycine (AVG; an inhibitor of ethylene production by inhibiting ACC synthesis) significantly increased, activities of the enzymes involved in the sucrose-to-starch conversion and the biosynthesis of amino acids, grain-filling rate, and amino acid content in the grain [76,77,78]; and (3) the adoption of moderate soil-drying treatment or an AWMD regime during grain filling significantly decreased the activities of ACC synthase and ACC oxidase, ACC content, and ethylene evolution rate, resulting in a greater grain-filling rate and higher content of amino acids in the ripe grain of rice and wheat [73,78,79,80]. In addition, ACC levels were observed to be significantly correlated with hydrogen peroxide (H_2_O_2_) content in plants, and a lower content of amino acids in rice grains was closely associated with higher levels of ACC and H_2_O_2_ in root exudates and grains [74,75,76,77,78,79,80,81], inferring that ethylene may impede amino acid biosynthesis through elevating ROS in the organ.

Since PAs (Spd and Spm) and ethylene share a biosynthetic precursor SAM (Figure 3), it is possible that increases in Spd and Spm biosynthesis could lead to the reduction in ethylene synthesis [67,68]. A potential metabolic interaction/competition between PAs (Spd and Spm) and ethylene biosynthesis may regulate the biosynthesis and metabolism of amino acids in the grain of rice. However, more direct evidence is needed to verify the roles of PAs and ethylene and their cross-talk in regulating the anabolism of amino acids in cereals.

### 3.2. Roles of BRs and Other Phytohormones

BRs are a unique class of naturally occurring plant polyhydroxylated steroid hormones and include brassinolide, castasterone, and their various derivatives, such as 24-epibrassinolide (24-EBL), 28-homobrassinolide (28-HBL), and 24-epicastasterone [82,83,84]. They play vital roles in plant growth and development, including seed germination, spikelet differentiation, pollen tube elongation, and flower and fruit production by regulating physiological and molecular processes, such as cell division, nucleic acid and protein biosynthesis, gene expression, and photosynthesis [85,86,87]. The recent work of Yang et al. [88] showed that compared with those in the grain of nontransgenic rice (a *Japonica* rice variety of nipponbare, wild type), increases in the contents of proteins, total amino acids, and Lys in the grains of transgenic rice (overexpression of genes of encoding Gln synthase *GS1*, nipponbare-*GS1*) were accompanied with increases in the contents of 24-EBL and 28-HBL, while there was no significant difference in the contents of IAA, cytokinins (zeatin and zeatin riboside) and gibberellins (GA_1_, GA_4_) between the two genotypes. Spraying a low concentration (10 µmol L^−1^) or high concentration (50 µmol L^−1^) of 24-EBL on rice panicles of Yangdao 6 (an *Indica* inbred) and Wuyunjing 24 (a *Japonica* inbred) at the early grain-filling stage significantly increased AK activity and contents of total amino acids and Lys in the grain compared with the control sprayed with distilled water. Spraying with a low concentration (10 µmol L^−1^) or high concentration (50 µmol L^−1^) of cytokinin (zeatin riboside), IAA, or gibberellin 3 (GA_3_) had no significant effect on AK activity in the filling grain and the contents of total amino acids and Lys in the ripe grain [88]. These results suggest that BRs play an important role in regulating amino acid biosynthesis in rice. The role of BRs in positively regulating amino acid biosynthesis was supported by the observations that the adoption of improved N management (N fertilizer was applied based on soil fertility, leaf color, and N requirement of a rice variety) or an AWMD regime significantly increased the contents of 24-EBL and 28-HBL in filling grains, which was concomitant with increases in the contents of essential amino acids, total amino acids, and Lys in ripe grains of both *Japonica* and *Indica* rice cultivars [73,89,90,91].

The mechanism by which BRs positively regulate the biosynthesis of amino acids in rice is not very clear. There are reports showing that BRs could increase energy charge by enhancing adenosine triphosphatase (ATPase, EC 3.6.1.3) activity, and that BRs could maintain redox homeostasis by enhancing the activities of enzymes involved in the ascorbate–glutathione (AsA-GSH) cycle, including glutathione reductase (EC 1.6.4.2) and monodehydroascorbate reductase (EC 1.6.5.4), and the expression of genes encoding these enzymes [91,92,93]. It is proposed that BRs can antagonize with ethylene, thereby minimizing the inhibitory effect of ethylene on amino acid biosynthesis in plants [93].

Abscisic acid (ABA) has also been observed to play an important role in mediating amino acid biosynthesis in rice [88,94]. It is reported that, under the stress of a high concentration of ammonium, the expression of genes related to ABA biosynthesis and the genes involved in the metabolism of flavonoids and some amino acids, such as proline (Pro), cysteine (Cys), and Met, were simultaneously upregulated in both *Japonica* and *Indica* rice seedlings [94]. In the grains of the transgenic rice line nipponbare-*GS1* (*Japonica*), a higher amino acid content was closely associated with a higher ABA content [88]. Spraying a low concentration (10 µmol L^−1^) of ABA on rice panicles at the early grain-filling stage significantly increased AK activity and the contents of total amino acids and Lys in the grains of both *Japonica* and *Indica* rice. However, spraying with a high concentration (50 µmol L^−1^) of ABA significantly reduced AK activity in the filling grains and the contents of total amino acids and Lys in the ripe grains. Furthermore, the inhibitory effect of spraying with high concentration of ABA on AK activity in the filling grains and the contents of total amino acids and Lys in the ripe grains was revised by the concomitant treatment of spraying with 10 µmol L^−1^ or 50 µmol L^−1^ 24-EBL [88]. These results suggest that there is an interaction between the BRs and ABA on amino acid synthesis, and that ABA may have a dose-dependent effect on regulating the biosynthesis of amino acids—that is, a low concentration could enhance amino acid biosynthesis, whereas a high concentration could inhibit it. The mechanism underlying ABA regulating amino acid biosynthesis remains elusive. It is speculated that ABA plays an important role in regulating the ratio of carbon (C) to N by modulating C and N metabolism and accordingly mediating the biosynthesis of amino acids in plants [80,81,82,88]. There are reports showing that, under well-watered or moderate soil-drying conditions, ABA in rice and wheat plants could antagonistically interact with ethylene to mitigate grain filling and amino acid biosynthesis in the grains [77,79,80,81].

There are a few reports showing that other phytohormones, such as cytokinins and auxin, may participate in mediating amino acid biosynthesis in plants [94,95,96]. It is reported that an increase in the content of amino acids under drought stress was closely associated with the decrease in contents of endogenous cytokinins and IAA in *Japonica* rice seedlings [95]. IAA is proposed to enhance the uptake and transport of amino acids in plants by upregulating the expression of the genes involved in amino acid transport [3,97,98]. It is noteworthy, however, that phytohormones function as signaling molecules that interact through a complex network to mediate plant growth and development including yield and quality formation [99], although their roles and mechanisms in regulating the anabolism of amino acids in crops are still yet to be fully understood.

## 4. Proposal for Further Research

### 4.1. Temporal and Spatial Distributions of Amino Acids in the Filling Grain

There are significant differences in the content and compositions of amino acids in different parts (such as aleurone and endosperm) of ripe grains of rice [36,37,38]. Little is known, however, about whether such differences are closely associated with the temporal (at different filling stages) and spatial (at different parts of a grain) distributions of the content and compositions of amino acids in a grain during the filling stage. Research is needed to unravel the temporal and spatial distributions of the content and compositions of amino acids in a filling grain after anthesis, the changes in Gln content in maternal tissues (leaves, sheaths, stems, etc.) during grain filling, and the changes in the activities of enzymes involved in amino acid anabolism, especially in Lys biosynthesis and metabolism including GS, GOGAT, GOT, GPT, AK, DHPs, LKR, and SDH in a filling grain, and their relationships with the protein content and components and the content and compositions of amino acids, especially essential amino acids such as Lys, in different parts of a ripe grain.

### 4.2. Cross-Talk of PAs and Phytohormones in Mediating the Anabolism of Amino Acids

So far, studies on the role of PAs, ethylene, BRs, ABA, or other phytohormones, especially on the cross-talk between or among phytohormones/plant growth regulators, in regulating amino acid anabolism have mostly been conducted using experiments with the exogenous application of relevant plant growth regulators. In such experiments, evidence for both synergistic and antagonistic interactions between or among phytohormones/plant growth regulators in the regulation has been raised because of the differences in genetic backgrounds, tissues or organs, growth and development stages, and growth conditions of plants [100,101]. It is generally proposed that the regulation of phytohormones on plant growth and development including crop quality is a highly complex process, in which the reception of one hormone usually triggers the synergistic or agonistic interplay with other hormones [11,17,18,93]. The knowledge on the interplay of PAs, ethylene, BRs, or with other phytohormones in crop plants in mediating the biosynthesis and metabolism of amino acids is very limited. Further investigations are necessary to elucidate the role of PAs and plant hormones and the cross-talk between or among phytohormones/plant growth regulators in mediating amino acid anabolism by means of the use of mutants or transgenic plants with an attenuated capacity to respond to or synthesize growth regulators.

### 4.3. Regulation Techniques for Promoting Biosynthesis of Amino Acids in the Grain

Water and N are the two largest input resources in crop production, and they are also two major factors regulating grain yield and quality including amino acid biosynthesis in rice [56,102,103]. However, the information as to how water and N management practices regulate the temporal and spatial distributions of the content and compositions of amino acids in filling grains is very limited. This merits investigating the effects of N management (the rate and the time of N fertilizer application) and irrigation regimes including continuous flooding, AWMD, and alternate wetting and severe soil drying (AWSD) on Gln content in maternal tissues and the activities of the key enzymes in the anabolism of amino acids, especially Lys, in different parts of a grain and in different grain-filling stages. Studies are required to understand phytohormones in response to water and N management and their regulatory mechanism underlying the anabolism of amino acids in filling grains. It should be emphasized that genetic improvement is the most important way to improve rice quality [104,105,106]. In recent years, a number of good-quality rice varieties or lines have been bred through molecular breeding technologies such as transgenic and gene-editing [106,107,108]. Further studies are needed to breed rice varieties or lines with a high content of amino acids, especially essential amino acids such as Lys, in the ripe grains though genetically optimizing the spatial and temporal distributions of the content and compositions of amino acids in the grain during the filling.

## 5. Concluding Remarks

Increasing the content of amino acids, especially Lys and other essential amino acids, in rice grains is of great significance to improve the nutritional and living standards of people in countries where rice is the staple food. PAs (Spd and Spm) and BRs (24-EBL and 28-HBL) play positive roles in mediating biosynthesis of amino acids in rice grains, mainly by enhancing the activities of the enzymes involved in amino acid biosynthesis and sucrose-to-starch conversion, upregulating the expression of the genes encoding these enzymes, promoting the uptake and transport of amino acids, maintaining redox homeostasis, and interacting antagonistically with ethylene (Figure 4). Further research is needed to elucidate the temporal and spatial distributions of the content and compositions of amino acids in the filling grains, and their relationships with the distributions of the content and compositions of amino acids in the ripe grain; to understand the role of phytohormones and plant growth regulators in mediating amino acid anabolism in the grains; and to establish regulation techniques for promoting the biosynthesis of amino acids in the grains by enhancing the balance and synergy between or among phytohormones and plant growth regulators.

## Figures and Tables

**Figure 1 plants-11-01581-f001:**
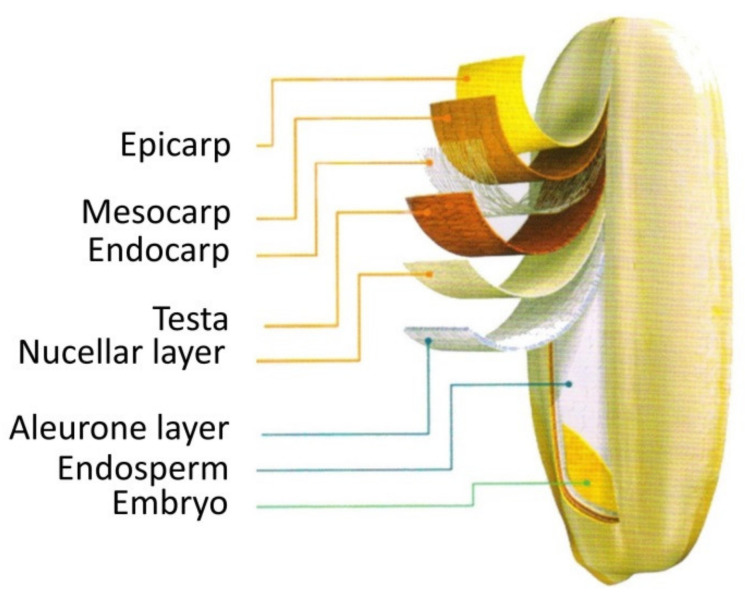
Morphology of a ripe rice grain (caryopsis). The figure is made according to [39,40].

**Figure 2 plants-11-01581-f002:**
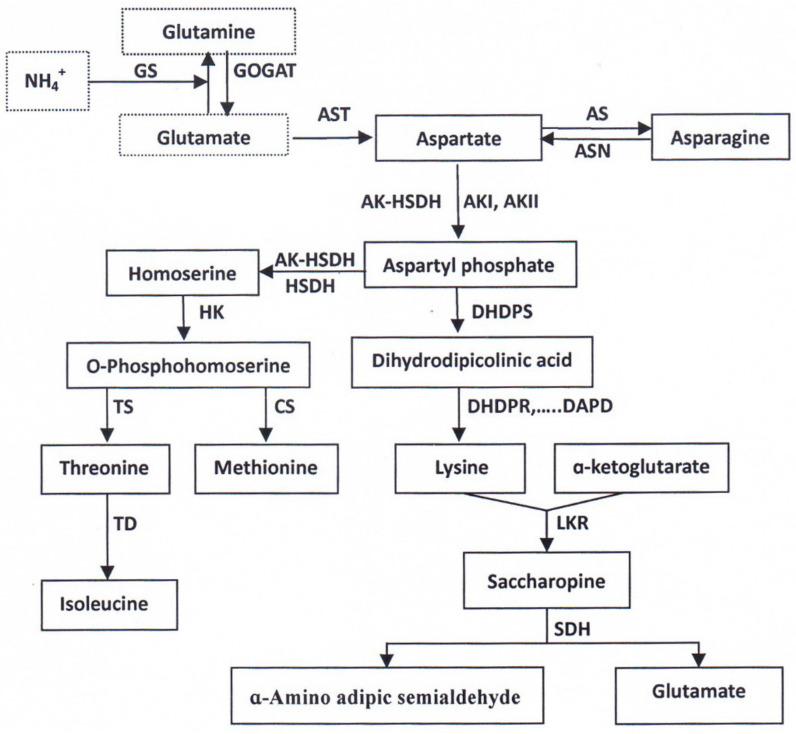
Amino acid metabolic pathway of aspartate family in higher plants. GS, glutamine synthetase; GOGAT, glutamate synthase; AST, aspartate transaminase; AS, asparagine synthetase; ASN, asparaginase; AK, aspartate kinase; HSDH, homoserine dehydrogenase; DHDPS, dihydropicolinate synthase; HK, homoserine kinase; TS, threonine synthase; CS, cysthationine synthase; TD, threonine dehydratase; DHDPR, dihydropicolinate reductase; DAPD, diaminopimelate decarboxylase; LKR, lysine α-ketoglutarate reductase; SDH, saccharopine dehydrogenase. The figure is made according to [42,59,60,61].

**Figure 3 plants-11-01581-f003:**
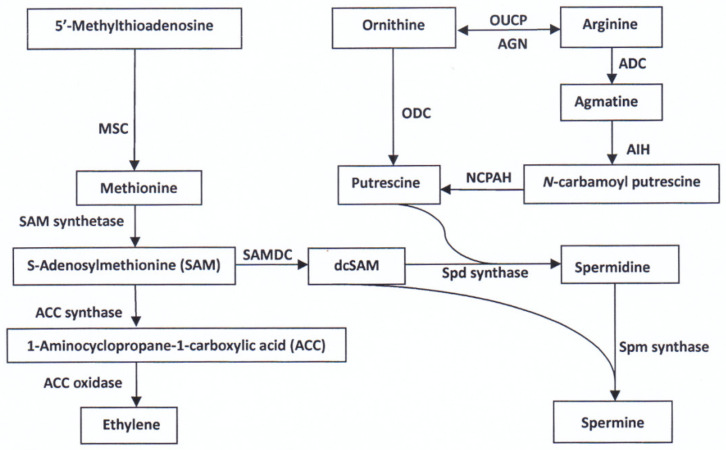
Principal pathways of polyamine and ethylene biosynthesis in plants. OUCP, ornithine–urea cycle complex; AGN, arginase; ADC, arginine decarboxylase; ODC, ornithine decarboxylase; MSC, methionine-synthesizing complex; AIH, agmatine iminohydrolase; NCPAH, *N*-carbamoylputrescine amidohydrolase; SAMDC, S-adenosylmethionine decarboxylase; dcSAM, decarboxylated S-adenosylmethionine; Spd, spermidine; Spm, spermine. The figure is made according to [66,67,68].

**Figure 4 plants-11-01581-f004:**
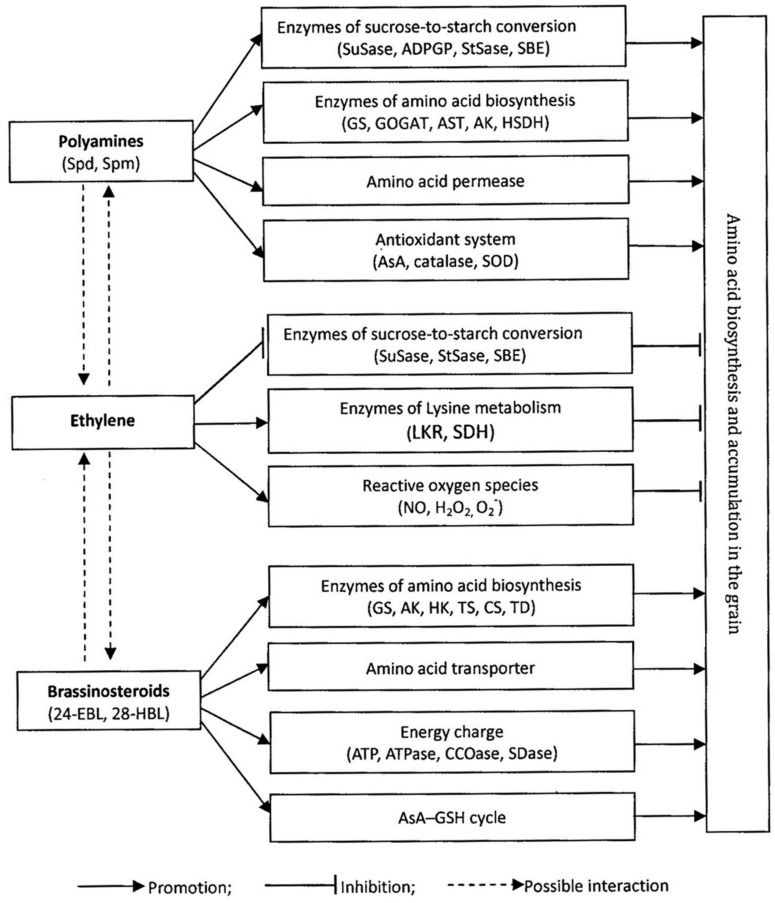
Schematic diagram of polyamines, ethylene, and brassinosteoids in mediating the biosynthesis of amino acids in the grain of rice. Spd, spermidine; Spm, spermine; 24-EBL, 24-epibrassinolide; 28-HBL, 28-homobrassinolide; SuSase, sucrose synthase; ADPGP, ADP glucose pyrophosphorylase; StSase, starch synthase; SBE, starch branching enzyme; GS, glutamine synthetase; GOGAT, glutamate synthase; AST, aspartate transaminase; AK, aspartate kinase; HSDH, homoserine dehydrogenase; AsA, ascorbic acid; SOD, superoxide dismutase; LKR, lysine α-ketoglutarate reductase; SDH, saccharopine dehydrogenase. HK, homoserine kinase; TS, threonine synthase; CS, cysthationine synthase; TD, threonine dehydratase; ATPase, adenosine triphosphatase; CCOase, cytochrome C oxidase; SDase, succinate dehydrogenase; AsA-GSH, ascorbate-glutathione. This diagram is assembled according to [62,63,64,65,66,68,69,70,71,72,88,89,90,91,92].

## Data Availability

The data presented in this review are adapted from relevant references.

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
