# Peer review of "Amino Acids in Rice Grains and Their Regulation by Polyamines and Phytohormones"

_plants, 2022, doi:10.3390/plants11121581_

Round 1
Reviewer 1 Report
The manuscript of review article Amino Acids in Rice Grains and their Regulation by Phytohormones” by Yang et al. submitted to Plants (1765662) refers to the important topic of amino acid composition and metabolism in rice grains.
The work is interesting and touches on an important topic, providing the human population with valuable food - rich in essential amino acids. The article contains mistakes, simplifications and should be revised before publication.
1. Introduction - the introduction is a copy of the abstract, it should focus more on outlining the aims of the paper.
2. Paragraph lines 73-93 authors should provide a figure a picture showing the structure – morphology/anatomy of rice seeds. Moreover, the authors should describe processing of rice grains, which will make the article easier to understand for people unfamiliar with cereal processing.
3. Line 95 “although the content of free amino acids in the ripe grain is very low,..” authors should provide values.
4. Line 115 wrong figure numbering, Fig 2 should be Fig 1.
5. Line 120 whether the authors mean free amino acids or amino acids in proteins?
6. Table 1. too detailed data - not like for a review article, the table should convey a simple message why the two varieties of rice do not have many differences between them? The data comes from a work in Chinese, it is not available to the international research community. Papers that are not in English should not be cited in a review.
7. Fig. 2 remarks as above.
8. Chapter “Roles of Phytohormones in Regulating Anabolism of Amino Acids in Rice”. The authors describe the effects of polyamine. Polyamines are not considered to be phytohormone. The authors should use the term growth regulators in relation to polyamines. Please rename the chapter – Phytohormones and Growth Regulators. The same remark applies to the title of the manuscript.
9. Line 296 Catalase is not a component of ascorbate-glutathione cycle!
10. Lines 301-304 rewrite the sentence.
11. Lines 355-359 different font.
12. The authors should point out the differences between Japonica and Indica rice. If the authors provide any information, they should indicate whether it is from Japonica or Indica rice.
13. References no. 2,24,27,29,38,46,48,50,51,66,70,71,85 are not written in English, they are not available to the international research community.
Author Response
Comments from the Reviewer 1
We should send million thanks to the Reviewer for the positive evaluation and very useful comments and suggestions. We have carefully considered the Reviewer’s comments and suggestions and revised the manuscript accordingly.
- Comment (C): Introduction - the introduction is a copy of the abstract, it should focus more on outlining the aims of the paper.
Response (R): We have taken the Reviewer’s suggestion and revised the section focused more on outlining and the aims of the paper.
- C: Paragraph lines 73-93 authors should provide a figure a picture showing the structure – morphology/anatomy of rice seeds. Moreover, the authors should describe processing of rice grains, which will make the article easier to understand for people unfamiliar with cereal processing.
R: We have taken the Reviewer’s suggestion and added the morphology of a ripe rice grain (caryopsis) as Figure 1. We also have simply decribed the processing of rice grains: “To improve palatability and digestion of rice, ripe grains (caryopses) need processing, in which the pericarp, testa, nucellar layer, aleurone layer, and embryo are removed by a polishing machine.”
- C: Line 95 “although the content of free amino acids in the ripe grain is very low,..” authors should provide values.
R: We have done it. “Although the content of free amino acids in the ripe grain is very low, approximately 0.35%-0.55% of the total amino acids”,…
- C: Line 115 wrong figure numbering, Fig 2 should be Fig 1.
R: We have corrected it.
- C: Line 120 whether the authors mean free amino acids or amino acids in proteins?
R: We have deleted the sentence.
- C: Table 1. too detailed data - not like for a review article, the table should convey a simple message why the two varieties of rice do not have many differences between them? The data comes from a work in Chinese, it is not available to the international research community. Papers that are not in English should not be cited in a review.
R: We have deleted Table 1 and the cited paper in Chinese.
- C: Fig. 2 remarks as above.
R: We have taken the Reviewer’s suggestion and deleted Figure 2.
- C: Chapter “Roles of Phytohormones in Regulating Anabolism of Amino Acids in Rice”. The authors describe the effects of polyamine. Polyamines are not considered to be phytohormone. The authors should use the term growth regulators in relation to polyamines. Please rename the chapter – Phytohormones and Growth Regulators. The same remark applies to the title of the manuscript.
R: We have taken the Reviewer’s suggestion and revised the Title and Chapter: Title: “Amino Acids in Rice Grains and Their Regulation by Polyamines and Phytohormones”. The Chapter: “Roles of PAs and Phytohormones in Regulating Anabolism of Amino Acids in Rice”.
- C: Line 296 Catalase is not a component of ascorbate-glutathione cycle!
R: Thanks for the Reviewer’s correction. We have deleted the description of catalase.
- C: Lines 301-304 rewrite the sentence
R: We have revised the sentence and asked MDPI to have edited English (English Editing ID: english-45682).
- C: Lines 355-359 different font.
R: We have corrected them.
- C: The authors should point out the differences between Japonica and Indica rice. If the authors provide any information, they should indicate whether it is from Japonica or Indica rice.
R: In the revised manuscript, we have indicated whether the data are from Japonica or from Indica rice. We have carefully read the papers cited in the manuscript and found that very few papers have made comparison between Japonica and Indica rice.
- C: References no. 2,24,27,29,38,46,48,50,51,66,70,71,85 are not written in English, they are not available to the international research community.
R: We have taken the Reviewer’s suggestion and deleted 12 references from the 13 references in non-English. The reference 85 (88 in the revised manuscript) is kept for two reasons: One is that this reference provides novel information about the role of brassinosteroids in regulating amino acid biosynthesis. Another is that the journal (Acta Agronomica Sinica) is very famous in Asia, and the abstract, tables and figures in the papers of the journal are written in English, and the journal has been collected by many international databases, such as FAO AGRIS, Biological Abstract (U.K.), Cambridge Scientific Abstract (U.S.A.), CAB Abstract (U.K),, Global Health (U.K.), Chemical Abstract (U.S.A), and Scopus (Netherlands).
We feel so grateful to the reviewer for so much tedious work done for us. Thanks again!
Reviewer 2 Report
I find the work interesting, scientifically sound, and timely, but I would suggest some reorganisation of it. I think that the concluding remarks should not include works cited. The works on which the conclusions are based should be reviewed in e main body of the paper.
Author Response
Comments from the Reviewer 2
Comments: I find the work interesting, scientifically sound, and timely, but I would suggest some reorganisation of it. I think that the concluding remarks should not include works cited. The works on which the conclusions are based should be reviewed in the main body of the paper.
Responses: We should send million thanks to the Reviewer for the positive evaluation and very useful comments and suggestions. We have taken the Reviewer’s suggestion and reorganized the concluding remarks, and divided the concluding remarks into two parts: the section 4: Proposal for Further Research which is the part of the main body, and the section 5: Concluding Remarks. This section has been much shorten (184 words).
Thanks again for the Reviewer’s kind comments and suggestions!
Round 2
Reviewer 1 Report
Dear Authors,
Thank you very much for considering my comments. I think the manuscript is now ready for publication.